# Conversion of the Propane–Butane Fraction into Arenes on MFI Zeolites Modified by Zinc Oxide and Activated by Low-Temperature Plasma

**DOI:** 10.3390/molecules25112704

**Published:** 2020-06-11

**Authors:** Vladimir I. Erofeev, Sofiya N. Dzhalilova, Mikhail V. Erofeev, Vasilii S. Ripenko, Vladimir P. Reschetilowski

**Affiliations:** 1School of Earth Sciences & Engineering, National Research Tomsk Polytechnic University, Tomsk 634050, Russia; dzhalilovasn@mail.ru; 2Institute of High Current Electronics SB RAS, Tomsk 634055, Russia; michael@loi.hcei.tsc.ru (M.V.E.); vripenko@loi.hcei.tsc.ru (V.S.R.); 3Technical University of Dresden, 01062 Dresden, Germany; wladimir.reschetilowski@tu-dresden.de

**Keywords:** MFI zeolite, modification, zinc oxide, plasma activation, acidity, propane–butane fraction, conversion, arenes

## Abstract

The effect of modification of MFI zeolite 1–5 wt.% ZnO activated by plasma on acid and catalytic properties in the conversion of the propane–butane fraction into arenes was investigated. The high-silica zeolites with silicate module 45 were synthesized from alkaline alumina–silica gels in the presence of an ‘X-oil’ organic structure-forming additive. The modification of the zeolite with zinc was carried out by impregnating the zeolite granules in the H-form with an aqueous solution of Zn(NO_3_)_2_. The obtained zeolites were characterized by X-ray phase analysis and IR spectroscopy. It is shown that the synthesized zeolites belong to the high-silica MFI zeolites. The study of microporous zeolite-containing catalysts during the conversion of C_3_-C_4_ alkanes to aromatic hydrocarbons made it possible to establish that the highest yield of aromatic hydrocarbons is observed on zeolite catalysts modified with 1 and 3% ZnO and amount to 63.7 and 64.4% at 600 °C, respectively, which is 7.7–8.4% more than on the original zeolite. The preliminary activation of microporous zeolites modified with 1–5% ZnO and plasma leads to an increase in the yield of aromatic hydrocarbons from the propane–butane fraction; the maximum yield of arenes is observed in zeolite catalysts modified with 1 and 3% ZnO and activated by plasma, amounting to 64.9 and 65.5% at 600 °C, respectively, which is 8.9–9.5% more than on the initial zeolite. The activity of the zeolite catalysts modified by ZnO and activated by plasma show good agreement with their acid properties. Activation of the zeolites modified by 1 and 3% ZnO and plasma leads to an increase in the concentration of the weak acid sites of the catalyst to 707 and 764 mmol/g in comparison with plasma-inactivated 1 and 3% ZnO/ZKE-XM catalysts at 626 and 572 mmol/g, respectively.

## 1. Introduction

The propane–butane fraction (PBF) is one of the main components of natural, associated petroleum gases and is a valuable organic raw material for the production of C_2_-C_4_ alkenes, arenes, and other valuable products. The most promising for PBF conversion to alkenes, arenes can be the MFI zeolite catalysts, which demonstrate high activity and selectivity in dehydrogenation, cracking, oligomerization, isomerization, and dehydrocyclization reactions of various organic compounds [1,2].

Previously, in most studies, it has been shown that the formation of the microporous MFI zeolites and their acid and catalytic properties are greatly influenced by the use of structure-forming additives, silicate module, synthesis conditions, and other factors. It is generally accepted that the activity and selectivity of the microporous MFI zeolites determine their acid properties and the presence and concentration of Lewis and Bronsted acid sites, which can be adjusted in various ways [3,4].

It was shown in references [5,6] that in the process of conversion of lower C_1_-C_6_ alkanes into liquid hydrocarbons, the high-silica ZSM-5 or MFI zeolites modified by gallium, zinc, molybdenum, and rhenium are the most active and selective. For example, the maximum yield of liquid hydrocarbons in the process of propane conversion reached 56 wt.% at 550 °C on the 3% Zn/ZSM-5 catalyst. It was also found that the synthesis methods of the high-silica ZSM-5 zeolites and the methods of introducing modifying additives into the zeolite and their nature have a great influence on the yield and selectivity of the formation of C_2_-C_4_ alkenes—arenes from lower alkanes [7,8,9].

In references [9,10], the influence of Zn/HZSM-5 synthesis methods on the formation of acid sites and their catalytic properties was investigated. It was found that in Zn-ZSM-5 synthesized by mechanical mixing of ZnO powders with zeolite, zinc is present in Zn-ZSM-5 mainly in the form of ZnO and ZnOH^+^ particles. In the Zn-ZSM-5 catalysts synthesized by direct hydrothermal ion exchange synthesis, zinc is present primarily in the form of ZnOH^+^. The Zn-ZSM-5 catalysts synthesized by the impregnation method mainly contain ZnOH^+^ and ZnO nanoparticles dispersed in the pores and channels of the zeolite. Thus, depending on the synthesis methods of Zn-ZSM-5 catalysts, zinc is found in the zeolites in various surface oxygen-containing thermostable structures. It has also been shown that cracking, dehydrogenation, and aromatization of light hydrocarbons into aromatics and lower alkanes predominantly occur at ZnOH^+^ acid sites.

In references [11,12], the authors studied the conversion of bioethanol and PBF on the microporous high-silica MFI zeolites, synthesized using multicomponent structure-forming additives of the “alcohol fraction” and the “X-oil” fraction (by-products of caprolactam industrial production) and their high catalytic activity was established in the conversion of ethanol and lower C_3_-C_4_ alkanes to liquid hydrocarbons and aromatic hydrocarbons, respectively.

The aim of this work was to study the physicochemical and catalytic properties of the microporous high-silica MFI zeolite synthesized using a new multicomponent structure-forming additive of the “X-oil” fraction (a by-product of caprolactam industrial production), modified by 1–5 wt.% ZnO and activated by low-temperature plasma, in the process of converting PBF-associated petroleum gases into arenes.

## 2. Results and Discussion

### 2.1. Synthesis and Characterization of Catalysts

The high-silica zeolite with silicate module 45 was synthesized from alkaline alumina–silica gels using a multicomponent organic structure-forming additive of the “X-oil” fraction (a by-product of caprolactam production) by hydrothermal synthesis in stainless steel autoclaves at 170–175 °C for 4–5 days (ZKE-XM), according to the procedure described in reference [13].

The specific surface of decationized zeolites in the H-form H-ZKE-XM modified with 1–5% ZnO was determined using the BET multipoint method on a Soft Sorbi, ver. 4.2. The specific surface area of zeolite H-ZKE-XM is 376 m^2^/g, and zeolite modification with zinc from 1 to 5 wt.% leads to an increase in the content of zinc oxide with a decrease in the specific surface of the modified zeolites. Thus, for example, the specific surface area of the catalyst 1% ZnO/99% H-ZKE-XM is 354 m^2^/g; 3% ZnO/97% H-ZKE-XM—346 m^2^/g; 5% ZnO/95% H-ZKE-XM—328 m^2^/g. The change in the specific surface area of the modified zeolites can be explained by the fact that the ZnO particles formed during calcination at 600 °C are located mainly in the pores, channels and pore mouths of the catalyst during preparation (impregnation of zeolite granules with a solution of zinc nitrate, then drying and calcination at 600 °C for 6–8 h). The internal surface of the zeolite is partially blocked and the specific surface of the modified zeolites decreases with an increase in the ZnO content from 1 to 5% in the pores and channels of the zeolite.

The physicochemical properties of zeolites modified with 1–5% ZnO were studied using X-ray phase analysis, IR spectroscopy and thermogravimetric analysis. X-ray patterns of zeolite H-ZKE-XM show lines with interplanar spacings (d. Å)—11.31; 10.12; 6.03; 4.37; 4.23; 4.07; 3.85; 3.70; 3.65; 3.36; 2.98; 1.99—which are typical for the MFI zeolite (Figure 1) [14,15]. X-ray diffraction patterns of 1–5% ZnO/99–95% H-ZKE-XM show lines with interplanar distances (d. Å)—11.05, 10.00, 6.76, 6.03, 5.72, 5.59, 4.63, 4.40, 4.29, 4.07, 4.02, 3.87, 3.83, 3.73, 3.66, 3.32, 3.06, 2.99, 2.49, 2.02—typical of the MFI zeolite (Figure 2).

Intense bands in the IR spectrum of N-ZKE-XM are observed at 451, 541, and 794 cm^−1^ with a wide band at 900–1300 cm^−1^ (Figure 3). The IR spectra of 1 and 3% ZnO/1–3% H-ZKE-XM are very similar to the IR spectrum of pure H-ZKE-XM.

The absorption band at 541 cm^−1^ corresponds to five membered rings in the zeolite framework and is typical for the ZSM-5 zeolites, or according to the new zeolite classification, for the MFI zeolites. The strong absorption band at 900–1300 cm^−1^ corresponds to antisymmetric stretching vibrations of TO_4_ tetrahedra, and the absorption band at 794 cm^−1^ refers to stretching vibrations of SiO_4_ tetrahedra.

Due to the low ZnO content (not more than 5%) and taking into account the large surface of zeolites, the ZnO bulk phase does not appear in modified zeolite catalysts in the IR spectra and in X-ray diffraction patterns. It is possible that zinc oxide particles are in a highly dispersed state in the pores and channels of the zeolite and form additional active centers of the catalyst with active centers of the zeolite. Based on the results of IR spectroscopy and X-ray phase analysis, we can conclude that the synthesized zeolites are microporous zeolites of the MFI type (ZSM-5) [14].

### 2.2. Catalytic Activity

The studies of the PBF conversion process on the initial H-ZKE-XM zeolite showed that with increasing temperature from 525 to 600 °C and the volume flow rate of PBF of 240 h^−1^, the yield of liquid products increases from 50.8 to 56.0% due to a rise in the conversion of PBF from 74.6 to 81.6% (Figure 4 and Figure 5, Table 1). The yield of the gaseous products of the PBF conversion with an increase in the reaction temperature decreases from 49.2 to 44.0%; the main products among gaseous hydrocarbons are methane and ethane.

The dependence of the PBF conversion on temperature on the zeolite catalysts is shown below:

With a growth of the reaction temperature of the gaseous products of PBF conversion, the yield of lower olefins (ethylene and propylene) rises, and the total content of C_2_-C_3_ olefins increases from 6.6 to 12.7% due to the intensification of dehydrogenation and cracking reactions’ lower alkanes C_3_-C_4_ [12,16,17].

The liquid PBF conversion products on the modified microporous zeolite catalysts consist of arenes: benzene, toluene, xylenes (BTX fraction) and methyl and ethyl naphthalenes. The yield of benzene with an increase in the conversion temperature of PBF from 525 to 600 °C increases from 11.9 to 14.7%. the total content of the BTX fraction changes slightly from 80.56 to 79.0% with increasing temperature from 525 to 600 °C (Figure 5, curve 1).

The main product among aromatic hydrocarbons is toluene, of which content with a rise in the conversion temperature of PBF is 40.0–42.2%. The total content of the xylenes and naphthalene derivatives decreases with the rise of the PBF conversion temperature due to the dealkylation reactions of xylenes and methylnaphthalenes.

The introduction of 1–5% zinc oxide (1–5% ZnO/99–95% H-CKE-XM) into the initial zeolite H-ZKE-XM leads to the yield decrease in gaseous hydrocarbons and the yield increase in liquid hydrocarbons (arenes) with a rise in the reaction temperature of the PBF conversion from 525 to 600 °C (Figure 5, curves 2–4).

The main products of the conversion of PBF on microporous zeolite-containing catalysts among gaseous hydrocarbons are methane and ethane, and their total yield is 55–58%. The yield of gaseous products with an increase in the temperature of the PBF conversion reaction from 525 to 600 °C decreases for all samples of H-ZKE-XM modified with 1–5% ZnO. With an increase in the temperature of the PBF conversion process from 525 to 600 °C, the maximum decrease in the yield of gaseous products is observed from 47.1 to 35.6% for the zeolite-containing catalyst 3% ZnO/97% H-ZKE-XM.

The modification of the initial zeolite H-ZKE-XM with 1–5% zinc oxide (1–5% ZnO/99–95% H-ZKE-XM) leads to an increase in the yield of arenes with a rise in reaction temperature of the PBF conversion from 525 to 600 °C and the degree of the PBF conversion from 76.5 to 91.2% for 3% ZnO/97% H-ZKE-XM compared to the original H-ZKE-XM (Figure 4 and Figure 5, curves 2–4).

The highest yield of the liquid PBF conversion products is achieved at 600 °C on 1% ZnO/99% H-ZKE-XM and 3% ZnO/97% H-ZKE-XM catalysts and are 63.7% and 64.4%, respectively, at 600 °C and the volume flow rate of 240 h^−1^ (Figure 5, curves 2–4). It should be noted that the introduction of zinc oxide into the H-ZKE-XM zeolite from 1 to 5% leads to a rise in reaction temperature of the PBF conversion from 525 to 600 °C to an increase in the content of benzene in the liquid products to 24.0% at 600 °C for the 5% ZnO/95% H-ZKE-XM catalyst compared with 14.7% of benzene formed on the initial H-ZKE-XM at 600 °C. In addition to benzene, the BTX fraction is the main component among liquid products (Figure 6, curves 2–4). The maximum content of toluene among the BTX fraction is observed on the 3% ZnO/97% H-ZKE-XM catalyst at 525 °C and is 44.5%, with a further rise in the process temperature to 600 °C. The content of toluene and xylenes on all catalysts decreases due to the predominant dealkylation reactions of alkyl aromatic hydrocarbons.

Preliminary low-temperature plasma activation of zeolite-containing 1–5% ZnO/99–95% H-ZKE-XM catalysts leads to a significant increase in the degree of PBF conversion from 86.5 to 94.0% and the yield of liquid products with a rise in reaction temperature of the PBF conversion from 525 to 600 °C (arenes) compared to the initial H-ZKE-XM (Figure 4 and Figure 5, curves 5–7).

The maximum yield of the liquid PBF conversion products is 64.9–65.5% at 600 °C on 1 and 3% ZnO/99–97% H-ZKE-XM catalysts, which is significantly higher than on pure H-ZKE-XM (56.0%) at 600 °C and the volume flow rate of 240 h^−1^ (Table 1, Figure 5, curves 5–7). It is important to note that the modification of the H-ZKE-XM zeolite of 1–5% ZnO and the preliminary catalyst activation with a low-temperature plasma leads to a rise in reaction temperature of the PBF conversion from 525 to 600 °C, and to a significant increase in the benzene content in the liquid products of the PBF conversion to 26.0% on 1% ZnO/99% H-ZKE-XM at 600 °C, compared to 14.7% of benzene formed on the initial H-ZKE-XM at 600 °C. The main products among the liquid products of the PBF conversion on zeolite catalysts modified by 1–5% ZnO and subjected to preliminary low-temperature plasma activation are benzene, toluene, and xylenes, the contents of which are 18.6–26.0%, 38.0–44.4%, and 12.3–20.4%, respectively (Figure 5 and Figure 6, curves 5–7). The total content of the BTX fraction with an increase in the conversion temperature of PBF is likely to decrease due to intensive dealkylation reactions [18,19,20,21,22].

### 2.3. Thermogravimetric Study of Coked Zeolite-Containing Catalysts

Thermogravimetric studies of zeolite catalysts modified with ZnO showed that three peaks are observed on the DSC curve of sample H-ZKE-XM (Figure 7).

On the DSC curve of the H-ZKE-XM sample in the range 50–150 °C a small endo effect at 108 °C is observed due to the removal of weakly adsorbed water and hydrocarbon vapors and a loss in catalyst mass of 0.57%.

The second small exoeffect at 300 °C apparently is associated with the burning out of coke deposits (“loose”, less condensed coke) from the surface of the coked zeolite catalyst and is accompanied by a loss in the weight of the catalyst by 2.63%.

In the region of 450–650 °C, predominantly more “dense” coke burns out at 565.7 °C, which is located mainly in the mouths or wide pores of the zeolite catalyst with a catalyst mass loss of 6.45% and is accompanied by a strong exothermic effect on the DSC curve (Figure 7).

On the DSC curve of a Carbonized Zeolite Catalyst 1% ZnO/99% H-ZKE-XM in the range 50–120 °C, an endoeffect at 98 °C is observed due to the removal of weakly adsorbed water and hydrocarbon vapors and is accompanied by a loss in catalyst mass of 2.89% (Figure 8).

The second small exoeffect at 290 °C apparently is associated with the burning out of coke deposits (“loose”, less condensed coke) from the surface of the coked zeolite catalyst and is accompanied by a loss in the mass of the catalyst by 0.56%.

In the region of 500–620 °C, the predominantly more “dense” coke burns out at 569.5 °C, which is located mainly in the mouths or wide pores of the zeolite catalyst and is accompanied by a loss in catalyst mass of 1.53% and accompanied by an exothermic effect on the DSC curve (Figure 8).

On the DSC curve of a Carbonized Zeolite Catalyst 3% ZnO/97% H-ZKE-XM in the range 50–120 °C, an endoeffect at 91 °C is observed due to the removal of weakly adsorbed water and hydrocarbon vapors and is accompanied by a loss in catalyst mass of 4.19% (Figure 9).

The second exoeffect in the form of a shoulder at 300 °C is apparently associated with the burning out of coke deposits (“loose”, less condensed coke) from the surface of the coked zeolite catalyst. In the region of 500–700 °C, the predominantly more “dense” coke burns out at 568.5 °C and in the form of a shoulder at 620–630 °C located mainly in the mouths and pores of the zeolite catalyst and is accompanied by a loss in the mass of the catalyst of 3.74% on the DSC curve (Figure 9).

On the DSC curve of a Carbonized Zeolite Catalyst 3% ZnO/97% H-ZKE-XM previously subjected to plasma treatment in the range 50–110 °C, an endoeffect is observed at 98.5 °C due to the removal of weakly adsorbed water and hydrocarbon vapors and is accompanied by weight loss of 3.42% of the catalyst (Figure 10).

The second exoeffect comes in the form of a shoulder at 300 °C, apparently due to the burning out of coke deposits (“loose”, less condensed coke) from the surface of the coked zeolite catalyst and the loss of coke mass of 0.37%. In the range of 500–700 °C, a strong exothermic peak splits into 2 exothermic effects at 551.3 and 648.2 °C and is accompanied by a loss in catalyst mass of 1.11 and 1.0% mostly more “dense” coke burns out, which is located mainly in the mouths and pores of the zeolite catalyst (Figure 10).

Thus, a thermogravimetric study of coking zeolite catalysts modified with 1–5% ZnO and subjected to preliminary plasma treatment during the conversion of the propane–butane fraction to arenes showed that mainly 2 types are formed in the reaction of the conversion of the propane–butane fraction to arenes on modified zeolite catalysts coke deposits: less condensed so-called “loose coke” and fading at a temperature of 290–310 °C and more condensed “dense coke” fading at a higher temperature of 500–700 °C.

### 2.4. Study of the Acidic Properties of Microporous Zeolite Catalysts

The study of the acid properties of the zeolite catalysts modified by ZnO showed that the initial H-ZKE-XM has two forms of ammonia desorption: The low-temperature peak reaches the thermal desorption curve (TD) in the region of 100–300 °C with a maximum temperature (T_m_) and with the peak at 185 °C, and relates to ammonia desorption mainly from the weak Lewis acid sites of the zeolite, which are coordinately unsaturated aluminum ions in the zeolite crystal lattice in H-ZKE-XM. The high-temperature peak, in the region of 350–500 °C with the T_m_ peak at 405 °C, related to the desorption of ammonia with the strongly Bronsted acid sites of the zeolite hydrogen ions of which are bridged hydroxyl groups (Figure 11, curve 1, Table 2).

The concentration of weak acid sites for 1% ZnO/99% H-ZKE-XM determined by the amount of desorbed ammonia is 600, and for strong-acid sites, it is 421 μmol/g [12,23]. The modification of H-ZKE-XM 1–5% ZnO leads to a change in their acid characteristics. The low-temperature ammonia desorption peaks from H-ZKE-XM zeolites modified by 1–5% ZnO are shifted to higher temperatures and T_m_ peaks are 185 and 190 °C for 1% ZnO/99% H-ZKE-XM and 5% ZnO/95% H-ZKE-XM, respectively, compared with the initial H-ZKE-SF zeolite (Figure 11, Table 2).

Additionally, the high-temperature ammonia desorption peaks from H-ZKE-XM zeolites modified by 1–5% ZnO are shifted to higher temperatures from T_m_ peaks of 405 °C for H-ZKE-XM and up to 410 °C for H-ZKE-XM samples modified by 1–5% ZnO.

The concentration of weak acid sites for 1% ZnO/99% H-ZKE-XM increases to 626 μmol/g compared to 600 μmol/g for H-ZKE-XM, with a further increase in the content from 1 to 5% ZnO in H-ZKE-XM; the concentration of weak acid sites decreases from 626 to 450 μmol/g (Table 2). The concentration of the strong acid sites for H-ZKE-XM is 421 μmol/g, and with an increase in the content from 1 to 5% ZnO in H-ZKE-XM, the concentration of strong acid sites increases from 151 to 258 μmol/g (Table 2).

The acid characteristics for H-ZKE-XM modified by 1–5% ZnO and subjected to preliminary low-temperature plasma activation are significantly different from the acid characteristics of H-ZKE-XM modified by 1–5% ZnO. The low-temperature ammonia desorption peaks from the H-ZKE-XM zeolites modified by 1–5% ZnO and activated by plasma are strongly shifted to higher temperatures and are T_m_ peaks of 210 and 215 °C for 1% ZnO/99% H-ZKE-XM and 3% ZnO/99% H-ZKE-XM, respectively, compared with the initial H-ZKE-XM zeolite (Table 2). Additionally, the ammonia desorption high-temperature peaks from H-ZKE-XM zeolites modified by 1–5% ZnO and activated by plasma are significantly shifted to higher temperatures from the T_m_ peaks of 405 °C for H-ZKE-XM and to 455 and 465 °C for the H-ZKE-XM samples modified by 1 and 3% ZnO and activated by low-temperature plasma.

It is important to note that with an increase in the content from 1 to 5% ZnO in H-ZKE-XM and being subjected to preliminary low-temperature plasma activation, the concentrations of weak acid and strong acid sites increase from 707 to 783 and from 201 to 232 μmol/g, respectively (Table 2). The total concentration of H-ZKE-XM acid sites modified by 1–5% ZnO and subjected to plasma activation, with an increase in the concentration from 1 to 5% ZnO, significantly increases from 908 to 1015 μmol/g.

This behavior of increasing the concentration of both types of acid sites in the zeolites, modified by 1–5% ZnO and activated by plasma treatment, as well as a significant increase in the yield of liquid hydrocarbons from PBF from 56.0% for H-ZKE-XM to 65.5% for the 3% ZnO/97% H-ZKE-XM zeolite catalyst can be explained by the fact that during the preliminary low-temperature plasma activation, there is a strong ionization of oxygen molecules and water vapor that are present in the air of the plasma chamber, and the resulting O^2−^, OH^−^ and H^+^ ions interact with various acid sites of the catalyst, whereby they form additional weak acid and strong acid sites on the surface of activated and modified by the plasma zeolite catalysts, and this leads to a significant increase of activity and an increase in the yield of liquid hydrocarbons (arenes) during the conversion of propane–butane fraction.

## 3. Materials and Methods

### 3.1. Catalysts Preparation

Microporous high-silica zeolites with silicate module 45 were synthesized from alkaline alumina–silica gels using a multicomponent organic structure-forming additive of the “X-oil” fraction (a by-product of caprolactam production) by hydrothermal synthesis in stainless steel autoclaves at 170–175 °C for 4–5 days, according to the procedure described in reference [13]. The conversion of the zeolite to the H-form (H-ZKE-XM) was carried out by decationization. The Na-CKE-XM zeolites were converted into the active form of H-ZKE-XM by treatment with a 1 M aqueous solution of NH_4_NO_3_ at 90 °C and stirring for 2 h, then the zeolite precipitate was filtered off from the mother liquor, then dried for 2 h at 110 °C and calcined for 6 h at 600 °C.

After that, the zeolite powder was molded, a fraction of 2–3 mm was selected and it was modified by 1–5 wt.% ZnO using the impregnation method. The calculated amount of zinc nitrate was dissolved in water, and then, a certain amount of H-ZKE-XM zeolite granules was impregnated with an aqueous solution of zinc nitrate according to the moisture capacity of the zeolite. Then, the catalysts were dried and calcined for 6 h at 600 °C.

### 3.2. Activation of the H-ZKE-XM Zeolite Catalysts Modified by 1–5% ZnO, by Low Temperature Plasma

The activation of the H-ZKE-XM zeolite catalysts modified by 1–5% ZnO was carried out by a low-temperature diffuse discharge plasma formed in atmospheric pressure air at 4 nanosecond voltage pulse with an amplitude of 25 kV applied from an NPG-3500N generator to a point cathode with a small radius of curvature, equal to 0.2 mm [23]. The maximum electron concentration in the plasma of the diffuse discharge was 10^14^ cm^−1^, with an applied average power of 2 W.

This type of discharge was formed when high voltage pulses of nanosecond duration were applied to an electrode with a small radius of curvature. As a result, an increase in the electric field occurred and the electrons in the discharge gap began to go into runaway mode (the so-called runaway electrons). As they moved from the cathode to the anode due to collisions with other molecules, the runaway electrons formed avalanches of runaway electrons intersecting each other, thus realizing the pre-ionization of the discharge gap.

The samples of the zeolite-containing catalysts in a quartz cuvette were placed on the grounded anode of the gas discharge chamber at a distance of 8 mm from the cathode tip, after which they were subjected to plasma treatment for 10 min at a frequency of 100 Hz, which corresponded to 60,000 pulses.

During the low-temperature plasma treatment of the catalysts, oxygen molecules and water vapor were strongly ionized, which were present in the air of the plasma chamber with the formation of O^2−^, OH^−^ and H^+^ ions [23]. The resulting O^2−^, OH^−^ and H^+^ ions further interacted with various acid sites of the catalyst, resulting in the formation of additional weak and strong acid sites on the surface of modified and plasma-activated zeolite catalysts.

### 3.3. Characterization Techniques

The physical and chemical properties of the obtained zeolites were studied using X-ray phase analysis and IR spectroscopy. X-ray phase analysis of the synthesized zeolites was carried out on a DRON-3 X-ray apparatus; Mo-anode and Ni-filter were used (Figure 1 and Figure 2). The IR spectra of the zeolite in the H-form were recorded using a Nicolet 5700 Fourier IR spectrometer in the region of 400–2000 cm^−1^ (Figure 2). The specific surface area of zeolite catalysts modified with 1–5% ZnO was determined using the BET multipoint method on a SoftSorbi, Ver. 4.2.

### 3.4. Acid Properties of Catalysts

The acid properties of the zeolite catalysts modified by 1–5% ZnO and activated by plasma were studied at the thermal desorption installation by adsorption of ammonia in a helium carrier gas stream in the range of 50–650 °C, with a linear heating rate of 10 °C/min according to the procedure described in reference [12]. The concentration of acid sites (μmol/g catalyst) in the zeolite-containing catalysts was determined from the amount of ammonia in desorption peaks (forms), and the accuracy of determining the amount of the adsorbed ammonia by the gas chromatography method was 2.5%.

### 3.5. Catalytic Activity Test

The catalytic properties of microporous zeolite-containing catalysts were studied in a catalytic installation in a stainless steel flow reactor, with an internal diameter of 16 mm and a length of 200 mm [12]. Catalyst granules (2 × 3 mm, volume 6 cm^3^) were loaded into the reactor. The conversion of PBF composition: methane—0.3; ethane—3.0; propane—80.9; butanes—15.8 wt.% was carried out on zeolite-containing catalysts in the temperature range of 525–600 °C, at the volume flow rate of 240 h^−1^, atmospheric pressure, and test duration of 2 h at each specific reaction temperature. The initial PBF and the products of the PBF conversion were analyzed by gas chromatography using the “Chromatek-Crystal 5000M” gas chromatograph.

The analysis of gaseous hydrocarbons was carried out on a quartz glass packing column (length 3 m, internal diameter 3 mm) filled with 5% NaOH on Al_2_O_3_ (fraction 0.25–0.50 mm) using a thermal conductivity detector, with the carrier gas being helium.

The analysis of the liquid products of the PBF conversion was carried out on a quartz glass capillary column (30 m × 0.25 mm × 0.5 μm) with a stationary ZB-1 phase deposited on a flame-ionization detector. Quantitative analysis of gaseous and liquid hydrocarbons was carried out on a hardware–software complex based on a “Chromatek-Crystal 5000” gas chromatograph using the results-based analysis program “Chromatek-Analytic” [12]. The error in determining the gaseous and liquid hydrocarbons by the gas chromatographic method was ±2.5%.

## 4. Conclusions

A study of the activity of the modified H-ZKE-XM zeolite catalysts of the MFI type during the aromatization of lower C_3_-C_4_ alkanes of the propane–butane fraction of associated petroleum gases showed that the introduction of zinc oxide modifying additives into the zeolite in the amount of 1–5 wt.% significantly increased the yield of liquid products—aromatic hydrocarbons from PBF. The introduction of 1–5% ZnO into the H-ZKE-XM zeolite and subsequent plasma activation of zeolite catalysts can significantly increase the strength and concentration of acid sites of catalysts and thereby increase the yield of aromatic hydrocarbons to 64.9% in the liquid products of the PBF conversion reaction at 600 °C, which is 12.5% more than on the initial H-ZKE-XM and 2–8% more than on 1–3% ZnO/99–97% H-ZKE-XM. It was shown that the activity of the zeolite catalysts modified by oxide zinc and activated by low-temperature plasma is in good agreement with their acid properties. This behavior of the modified zeolite catalysts can apparently be explained by the fact that during the activation of the modified zeolite catalysts by low-temperature plasma, strong ionization of oxygen molecules and water vapors, which are present in the air of the plasma chamber. The resulting O^2−^, OH^−^ and H^+^ interact with various acid sites of the catalyst, resulting in the formation of additional weak and strong acid sites on the surface of the modified and active plasma zeolite catalysts, which leads to an increase in their activity in the reactions of dehydrogenation, cracking and dehydrocyclization of lower C_3_-C_4_ alkanes. In this regard, on zeolite catalysts modified by zinc oxide and activated by low temperature plasma, a higher yield of methane, ethane and aromatic hydrocarbons is observed during the conversion of lower C_3_-C_4_ propane alkanes of the propane–butane fraction, than on zeolites modified by zinc oxide.

## Figures and Tables

**Figure 1 molecules-25-02704-f001:**
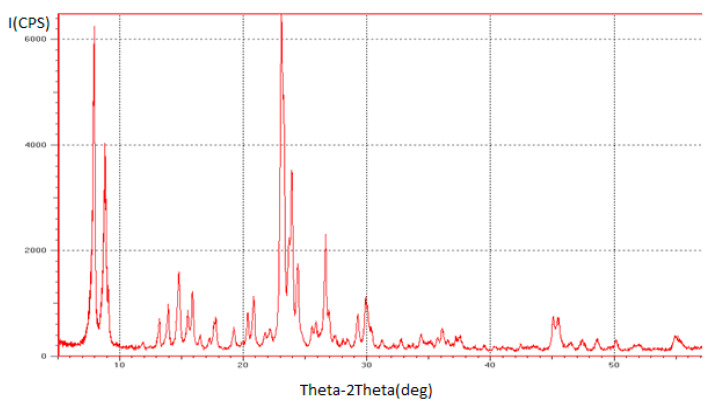
The X-ray diffraction pattern of the H-ZKE-XM zeolite catalyst.

**Figure 2 molecules-25-02704-f002:**
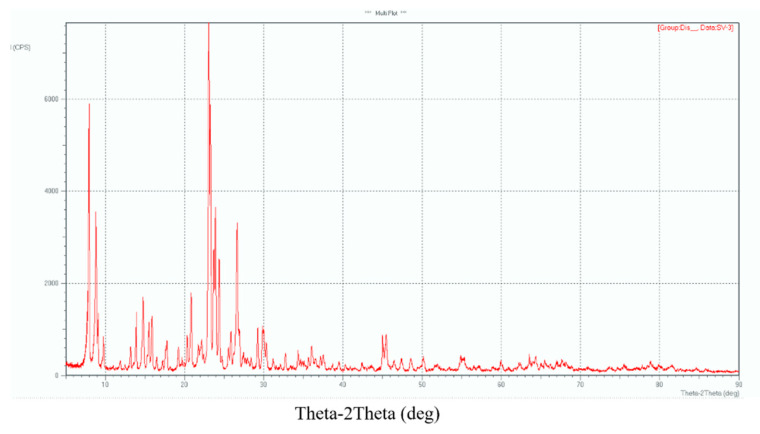
The X-ray diffraction pattern of the 3% ZnO/97% H-ZKE-XM zeolite catalyst.

**Figure 3 molecules-25-02704-f003:**
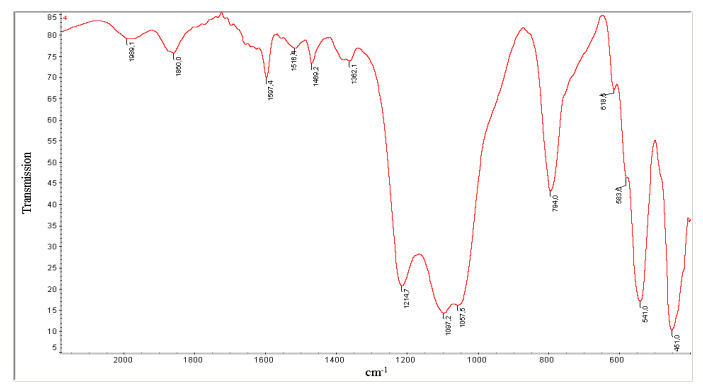
IR spectrum of the H-ZKE-XM zeolite catalyst.

**Figure 4 molecules-25-02704-f004:**
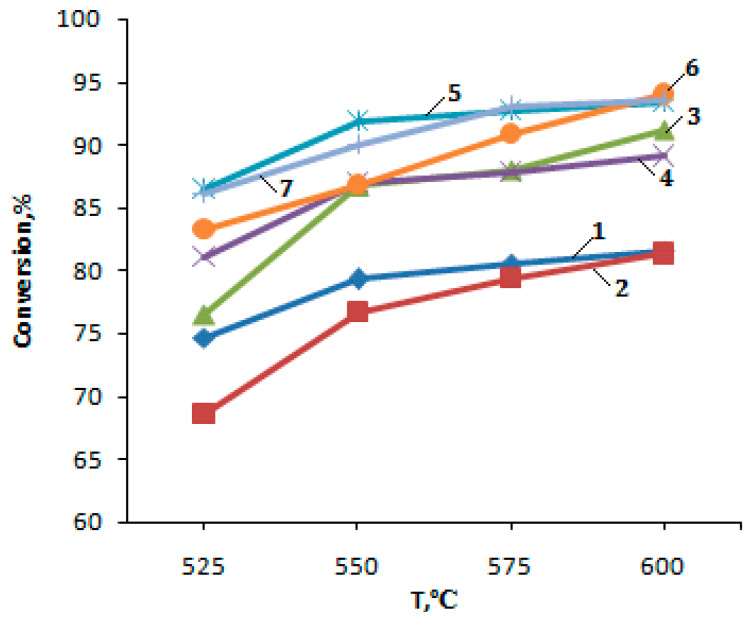
Dependence of the PBF conversion on temperature on the zeolite catalysts: 1—H-ZKE-XM; 2—1% ZnO/99% H-ZKE-XM; 3—3% ZnO/97% H-ZKE-XM; 4—5% ZnO/95% H-ZKE-XM; 5—1% ZnO/99% H-ZKE-XM (plasma); 6—3% ZnO/97% H-ZKE-XM (plasma); 7—5% ZnO/95% H-ZKE-XM (plasma).

**Figure 5 molecules-25-02704-f005:**
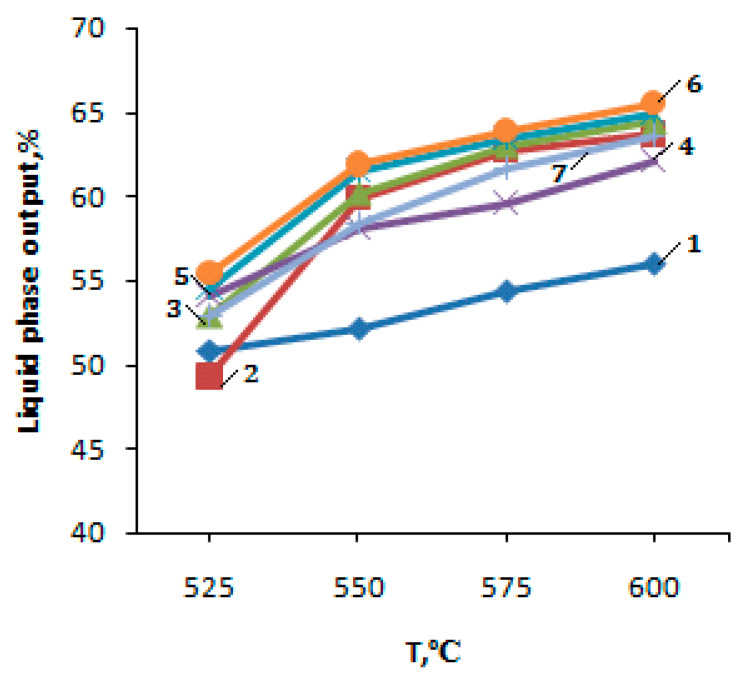
Dependence of the yield of arenes from PBF on temperature on the zeolite catalysts: 1—H-ZKE-XM; 2—1% ZnO/99% H-ZKE-XM; 3—3% ZnO/97% H-ZKE-XM; 4—5% ZnO/95% H-ZKE-XM; 5—1% ZnO/99% H-ZKE-XM (plasma); 6—3% ZnO/97% H-ZKE-XM (plasma); 7—5% ZnO/95% H-ZKE-XM (plasma).

**Figure 6 molecules-25-02704-f006:**
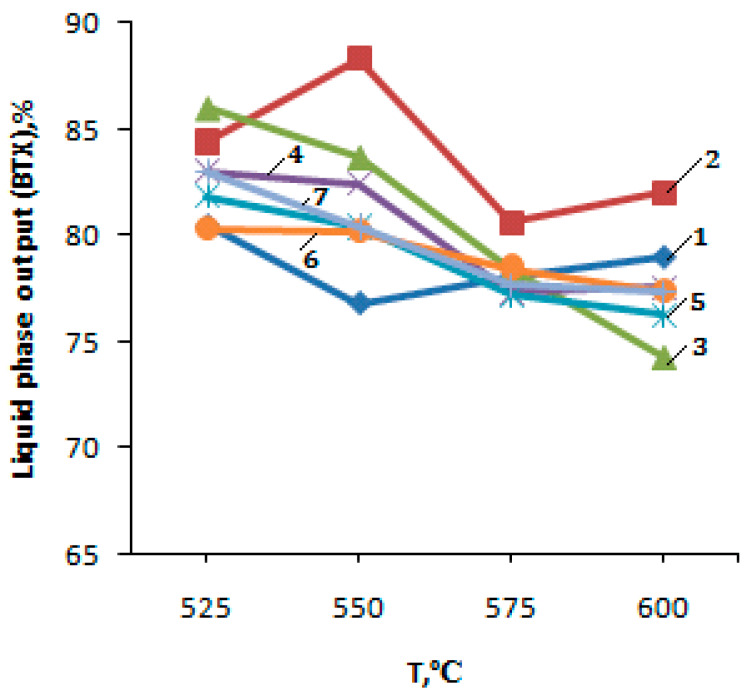
The dependence of the yield of arenes from the BTX fraction on temperature on the zeolite catalysts: 1—H-ZKE-XM; 2—1% ZnO/99% H-ZKE-XM; 3—3% ZnO/97% H-ZKE-XM; 4—5% ZnO/95% H-ZKE-XM; 5—1% ZnO/99% H-ZKE-XM (plasma); 6—3% ZnO/97% H-ZKE-XM (plasma); 7—5% ZnO/95% H-ZKE-XM (plasma).

**Figure 7 molecules-25-02704-f007:**
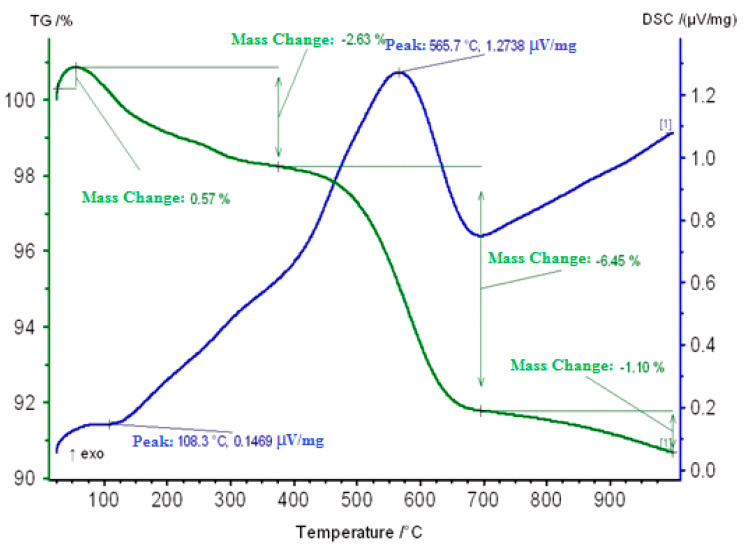
Thermogram of a Carbonized Zeolite Catalyst H-ZKE-XM.

**Figure 8 molecules-25-02704-f008:**
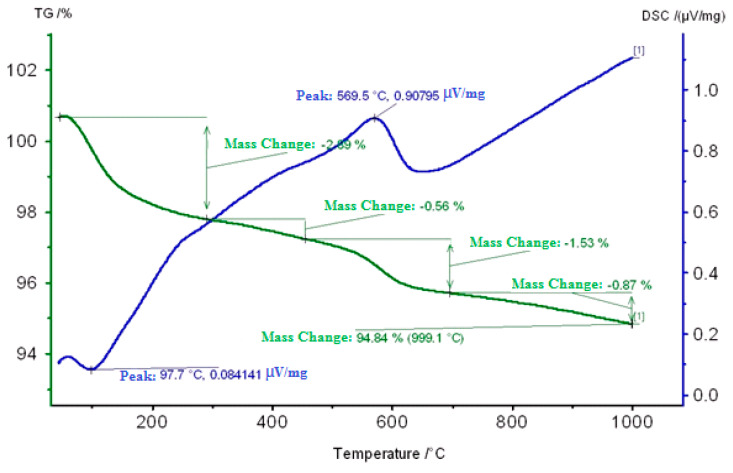
Thermogram of a Carbonized Zeolite Catalyst 1% ZnO/99% H-ZKE-XM.

**Figure 9 molecules-25-02704-f009:**
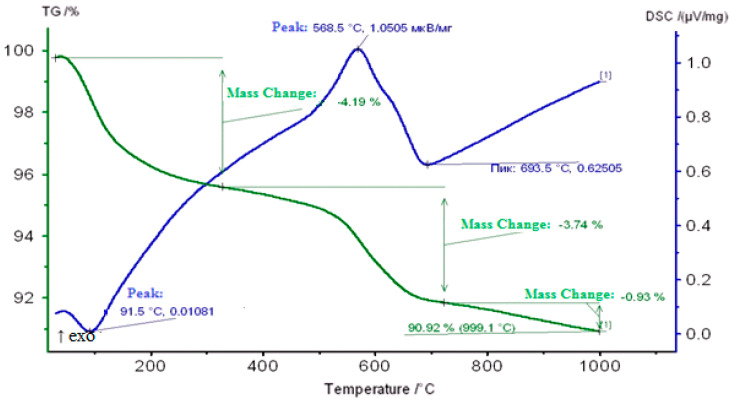
Thermogram of a Carbonized Zeolite Catalyst 3% ZnO/97% H-ZKE-XM.

**Figure 10 molecules-25-02704-f010:**
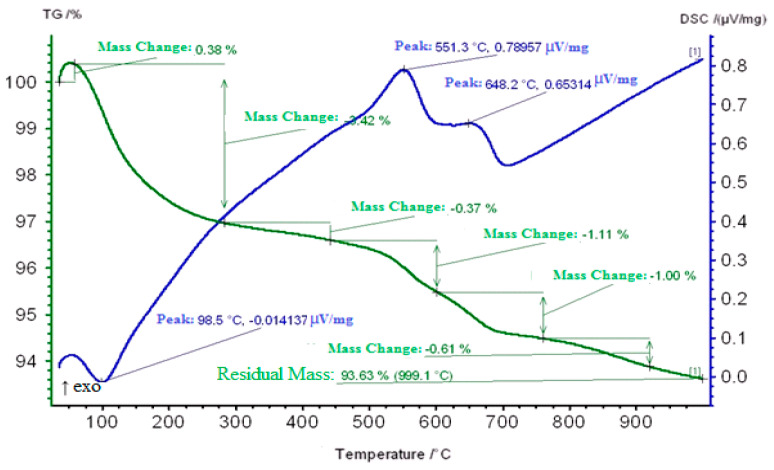
Thermogram of a Carbonized Zeolite Catalyst 3% ZnO/97% H-ZKE-XM (plasma).

**Figure 11 molecules-25-02704-f011:**
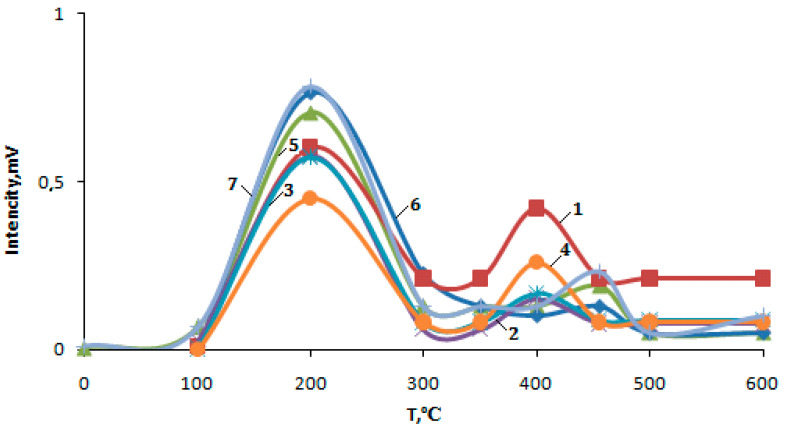
Thermal desorption spectrum of ammonia on H-ZKE-XM modified by 1–5% ZnO: 1—pure H-ZKE-XM. 2—1% ZnO additive. 3—3% ZnO additive. 4—5% ZnO additive. 5—1% ZnO additive (plasma). 6—3% ZnO additive (plasma). 7—5% ZnO additive (plasma).

**Table 1 molecules-25-02704-t001:** Conversion of the propane–butane fraction to arenes on zeolite catalysts modified with 1–5% ZnO and subjected to plasma pretreatment.

Temperature °C	Conversion %	Gas Phase %	Gas Phase Output %		Liquid Phase %	Liquid Phase Output %				
			Alkanes C_1_-C_4_	Alkenes C_2_-C_3_		C_6_H_6_	C_7_H_8_	C_8_H_10_	C_9_	C_10+_
H-ZKE-XM										
525	74.6	49.2	93.4	6.6	50.8	11.9	35.6	33.0	4.4	15.1
550	79.3	47.8	90.3	9.7	52.2	11.4	35.4	30.0	5.4	13.1
575	80.6	45.6	89.3	10.7	54.4	13.7	37.5	26.9	6.2	15.7
600	81.6	44	87.5	12.7	56.0	14.7	38.1	26.2	4.7	16.3
1% ZnO/H-ZKE-XM										
525	68.6	50.7	94.7	5.3	49.3	15.8	40.0	28.6	2.0	13.6
550	76.7	40.1	94.1	5.9	59.9	19.9	43.4	25.0	3.1	12.6
575	79.4	37.3	93.2	6.8	62.7	19.0	41.6	20.0	3.1	16.3
600	81.3	36.3	91.7	8.3	63.7	20.6	42.2	19.2	3.0	15.0
3% ZnO/H-ZKE-XM										
525	76.5	47.1	97.0	3.0	52.9	16.7	44.5	24.8	1.1	12.9
550	86.7	39.8	96.5	3.5	60.2	19.9	42.4	21.4	0.7	15.6
575	88.0	37.0	95.4	4.6	63.0	21.1	40.5	16.8	0.4	21.2
600	91.2	35.6	90.8	9.2	64.4	21.5	37.2	15.6	0.4	25.3
5% ZnO/H-ZKE-XM										
525	81.1	45.9	97.2	2.8	54.1	15.3	42.7	25.0	0.8	16.2
550	87.0	41.9	96.4	3.6	58.1	19.9	42.5	20.0	0.5	17.1
575	87.8	40.4	94.4	5.6	59.6	22.0	39.0	16.3	0.5	21.2
600	89.1	37.9	88.5	11.5	62.1	24.0	38.3	15.2	0.4	22.1
1% ZnO(plasma)/H-ZKE-XM										
525	86.5	45.5	95.6	4.4	54.5	18.6	43.2	20.0	1.3	16.9
550	91.9	38.5	94.6	5.4	61.5	21.6	41.5	17.3	0.9	18.7
575	92.7	36.4	93.0	7.0	63.5	23.3	40.2	13.7	0.8	22.0
600	93.4	35.1	88.8	11.2	64.9	26.0	38.0	12.3	0.5	23.2
3% ZnO (plasma)/H-ZKE-XM										
525	83.2	44.6	96.0	4.0	55.4	19.4	43.4	17.5	1.6	18.1
550	86.8	38.1	94.7	5.3	61.9	21.7	43.2	15.3	1.1	18.7
575	90.8	36.1	92.8	7.2	63.9	22.3	41.6	14.6	0.6	20.9
600	94.0	34.5	89.9	10.1	65.5	23.6	40.4	13.4	0.5	22.1
5% ZnO (plasma)/H-ZKE-XM										
525	86.1	47.1	96.9	3.1	52.9	18.2	44.4	20.4	0.5	16.5
550	90.0	41.6	96.2	3.8	58.4	22.5	41.5	16.4	0.4	19.2
575	93.0	38.3	94.2	5.8	61.7	24.4	39.0	14.2	0.4	22.0
600	93.6	36.4	88.8	11.2	63.6	24.2	40.0	13.1	0.4	22.3

**Table 2 molecules-25-02704-t002:** Acidic properties of zeolite-containing catalysts *.

Catalyst	T_max_. °C		Concentration of Acid Sites. µmol/g		
	T_I_	T_II_	C_I_	C_II_	C_total_
H-ZKE-XM	185	405	600	421	1021
1% ZnO/99% H-ZKE-XM	185	405	626	151	777
3% ZnO/97% H-ZKE-XM	190	410	572	167	739
5% ZnO/95% H-ZKE-XM	190	410	450	258	708
1% ZnO/99% H-ZKE-XM; plasma	210	455	707	201	908
3% ZnO/97% H-ZKE-XM; plasma	215	465	764	220	984
5% ZnO/95% H-ZKE-XM; plasma	220	465	783	232	1015

* T_1_ T_11_—peak temperatures for forms 1 and 11; C_1_, C_11_ and C_total_—concentration of acidic centers in forms (1), (11) and total, respectively.

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
