# Peer review of "Conversion of the Propane–Butane Fraction into Arenes on MFI Zeolites Modified by Zinc Oxide and Activated by Low-Temperature Plasma"

_molecules, 2020, doi:10.3390/molecules25112704_

Round 1

Reviewer 1 Report

Journal: Molecules (ISSN 1420-3049)

Manuscript ID molecules-800370

Manuscript title: Conversion of the Propane-Butane Fraction into arenes on MFI Zeolites Modified by Zinc Oxide and Activated by Low-Temperature Plasma.

In this manuscript, Еrofeev and coworkers studied the influence of modifying MFI zeolite by zinc oxide (1-5%) activated by plasma on the acidic properties and the catalytic behavior towards the conversion of the propane-butane fraction into arenes.

In most of the parts, the authors analyzed the data without giving an explanation of the obtained results. This manuscript is closer to a technical report than to a scientific article. 

The following points are raised:

Main article:

- The catalyst used was very poorly characterized. Only XRD and FTIR were shown while no clue on the particle size or morphology was mentioned. None of the following techniques were used: DLS, SEM, TEM, TGA. Data from those experiments must be provided to readers to understand the studied catalyst.

- How was the Zn content evaluated in the absence of ICP analyses?

- What is the size of zinc oxide particles? Are they localized on the surface, within the voids, or both inside and outside? A microscopic study (SEM, TEM) is required.

- How did zinc oxide influence the zeolite textural properties (nitrogen adsorption analyses are required)?

- Catalysis: why was the 3% modified catalyst more efficient than the 5%?

-Line246: how was the moisture capacity of zeolite calculated?

Supplementary information:

Pages 1-3 are in Russian. If the authors find these information valuable for readers, they should translate them into English.

Pages 4-18 are a copy of the manuscript; they should be all removed.

Author Response

All changes have been made according to the comments of reviewers.

Reviewer 2 Report

The work addresses issues of clear industrial interest and is generally well conducted. Despite this, a sense of incompleteness cannot be avoided. The aim of the work was to study the physico-chemical and catalytic properties of the ZFI-modified MFI zeolite samples and activated with a low temperature plasma [line 72] but, the mother zeolite H-ZKE-XM is the only one sample subjected to XRD and IR Analysis. Although I'm aware that the changes of the materials in question should not dramatically influence the crystallinity of a zeolite with such a high silicon / aluminum ratio, I still think that there should be XRD analyzes of samples at 3% ZnO and 3% ZnO (plasma). (at the limit referred to the 5% samples if the ZnO phase was not sufficiently visible in the 3% samples). Furthermore, there are no chemical analyzes to quantify the amount of ion exchange with ammonium on the MFI zeolite. Regarding the comparative evaluation of the acid and catalytic properties, why was the H-ZKE-XM zeolite without ZnO but treated with low temperature plasma not included in the list of samples? A further problem which in my opinion is rather limiting for the evaluation of a catalyst is the absence of the surface values ​​(BET) of the samples. The introduction of the ZnO phase should tend to reduce the specific surface while what the plasma treatment could tend to do? An increase in the specific surface area of ​​the zeolitic samples? The increase in Lewis acid sites could lead to thinking about this. It would be important to be able to evaluate the results obtained in Tab. 1 and the PBF conversion rates also with respect to the surface area values ​​of the various samples. As a last request, even two new tables could be introduced: in the first one, the global conversion percentages of the PBF and the liquid phase percentages obtained at 525 ° C and 600 ° C for the various samples could be introduced; in the second, the composition of the liquid phase between 525 ° C and 600 ° C for the various samples. The reading of these data inserted only in the text was rather heavy. Ultimately, I recommend publishing on Molecules only after addressing the following points:

1) Insert xrd of a sample modified with ZnO (3% or 5%) and of a sample modified with ZnO and activated with plasma (3% or 5%);

2) Evaluation of the specific surface areas of the samples;

3) Introduce one or more tables that can collect the results of the catalytic activities

4) Line [110], caption Figure 3, curve 3: replace "to" with the symbol "-"

5) Line [127], caption Figure 4, curve 3: replace "to" with the symbol "-"

6) Line[134]: "conversion of PBF to microporous zeolite ....", replace "in" with "on";

7) At various points in the text the value of 240 h ^ -1 is indicated for the volumetric flow rate. What volume unit are you referring to? ml?

8) Line [193]: 55% ZnO / 95% H-ZKE-XM, remove a 5;

9) Line [194]: (Figure 3, Table) must become (Figure 6, Table 1);

10) In table 1 the symbol "Ц" must be replaced with "Z".

Round 2

Reviewer 1 Report

In the second version of the manuscript, the authors introduced some modifications and clarifications, especially  BET and TG experiments.

Some of the points raised during the first round of revision were not answered, these points are raised again hereunder:

  • Chemical analyses are required to evaluate the Zn content. The proposed values are based on the quantities added during the impregnation and not to experimentally verified ones.
  • If technically/logistically possible, the authors should include microscopic studies to monitor the shape and size of these materials.
  • Explain clearly why was the 3% modified catalyst more efficient than the 5%? (most probably it is a question of dispersion).
  • Translate p1-3 in supplementary information or remove them. All manuscript and SI content should be in English so that they target all readers.
  • Remove pages 4-18 from the supplementary information, they are a copy of the manuscript, they do not bring any additional value or information to the manuscript.

Author Response

Responses to the comments of 1 reviewer

  1. In 1% ZnO / 99% X-ZKE-XM catalyst, the actual content of ZnO was 0.98%, and for 3% ZnO / 97% X-ZKE-XM - 2.98%.

The error of the discrepancy between the calculated and actual content of ZnO in the samples of the catalysts is 3%, which allows us to take the calculated values of the concentration of ZnO as the basis.

  1. Microscopic studies of the catalysts were not included in this article. The nature of modified zeolites of the ZSM-5 type by zinc, gallium, and others has been the subject of many works, some of which are indicated in the list of references.

  1. It is well known that with increasing concentration of the modifier, the activity and selectivity of the catalyst passes through a maximum at a certain concentration. We have previously shown for many modified zeolite catalysts that the most optimal concentration is 1-3%, with a further increase in the content of the modifying additive, the activity of the catalyst decreases.

  1. Figures 1-3 are best left in the article, because their translation into the additional information section will often force the reader to look for them and constantly refer to them.

  1. Please delete pages 4-18 from the additional information. We cannot find them.

Corrections in the text of the article are highlighted in red.

Reviewer 2 Report

Dear authors, first of all I would like to say that I really appreciate the implementations introduced in your work. The table that in my pdf has "escaped" from the text by moving to the end of the manuscript (which must be called Table 1) effectively summarizes the many data proposed and is certainly a valuable update to improve the readability of the work.

I also appreciate the DSC / TG analyzes concerning the regeneration of the coked catalysts. Without too many words, I recommend the publication of this work on Molecules after a minor revision to address the following points:

1) line 31- "leads to a increase" must be changed to "leads to an increase";

2) line 82- The name H-ZKE-XM is introduced for the first time without any kind of explanation. The reader can only imagine that we are talking about an hydrogenionic form of zeolite. Please put a short explanation or otherwise, a reference to paragraph 3.1.

3) line 115- You say that the ZnO phase is invisible in the IR analyzes but the Figure 3, by the caption is clearly relative to a sample that doesn't contain ZnO. Please, clarifay that you haven't reported the IR graph of the sample contening the zinc oxide just because very similar to that present in Figure 3.

4) line153- As requested in the previous review, "The main products of the conversion of PBF to microporous zeolite" must be changed to "The main products of the conversion of PBF on microporous zeolite".

5) Table 2- The "Ц" letter is still present in the name of the first sample.

Regards

Author Response

Responses to the comments of 2 reviewer

  1. Line 31 - "in" is replaced by "on".

  1. Line 82 - the designation has added the designation "decationized zeolites in the H-form of H-ZKE-XM" '(line 82-83).

  1. 3. Line 115 – Added: The IR spectra of 1 and 3% ZnO/1-3% H-ZKE-XM are very similar to the IR spectrum of pure H-ZKE-XM. Due to the low ZnO content (not more than 5%) and taking into account the large surface of zeolites, the ZnO bulk phase does not appear in modified zeolite catalysts in the IR spectra and in X-ray diffraction patterns. It is possible that zinc oxide particles are in a highly dispersed state in the pores and channels of the zeolite and form additional active centers of the catalyst with active centers of the zeolite.

  1. Line 153 - "in" is replaced by "on".

Corrections in the text of the article are highlighted in red.

If the editors consider it necessary to edit the English language of the article, then we do not mind, please include the payment in the price of the article.

This manuscript is a resubmission of an earlier submission. The following is a list of the peer review reports and author responses from that submission.